# Well-Being of Orthodontic Patients Wearing Orthodontic Appliances

**DOI:** 10.3390/medicina60081287

**Published:** 2024-08-09

**Authors:** Rugilė Nedzinskaitė, Benedikta Augytė, Dalia Smailiene, Arūnas Vasiliauskas, Kristina Lopatiene, Egle Zasčiurinskiene, Giedre Trakiniene

**Affiliations:** Department of Orthodontics, Lithuanian University of Health Sciences, LT-50106 Kaunas, Lithuania; rugile.nedzinskaite@stud.lsmu.lt (R.N.); benedikta.augyte@stud.lsmu.lt (B.A.); dalia.smailiene@lsmu.lt (D.S.); arunas.vasiliauskas@lsmu.lt (A.V.); kristina.lopatiene@lsmu.lt (K.L.); egle.zasciurinskiene@lsmu.lt (E.Z.)

**Keywords:** health-related quality of life, orthodontic appliances, malocclusion

## Abstract

*Background and Objectives*: Orthodontic appliances may cause discomfort for patients. This could influence the person’s psychological well-being. The aim of this study was to examine the psychological health of patients wearing orthodontic appliances. It is important to analyze the well-being of patients during orthodontic treatment because it is started in a young age and it may affect one’s psychological health. Therefore, for doctors and patients, it is important to understand and take every aspect of the treatment into consideration when deciding when to start the treatment and what appliance to choose. *Materials and Methods*: A total of 339 patients filled out an anonymous questionnaire. They were divided into four groups according to their age and type of appliance. The main reasons to seek treatment were crooked teeth and a bad bite. There was a statistically significant difference between women and men as women sought treatment because of crooked teeth; meanwhile, men indicated bad bite as their main reason for seeking treatment. *Results*: A total of 48.7% of subjects noted that they were feeling good during treatment; however, 4% of all patients noted that they were feeling bad. A total of 45% of subjects stated that they felt unhappy at least sometimes. A total of 43% of subjects felt stress. *Conclusions*: Even though patients indicated various negative aspects on their well-being during treatment, for the majority of the subjects, the end results significantly increased their psychological well-being.

## 1. Introduction

The World Health Organization’s (WHO’s) constitution states that health is a state of complete physical, mental and social well-being and not merely the absence of disease or infirmity [1]. Mental health is an integral and essential component of health [1]. Thus, a healthy oral cavity is one of the components of one’s well-being which for orthodontic patients depends mainly on the comfort while wearing orthodontic appliances.

According to the WHO, tooth decay, periodontological diseases and formations of malocclusions are three most commonly found disorders of the oral cavity [1]. As they are such common problems, a lot of countries are investigating the prevalence of these anomalies [2]. It was found that the most common malocclusion was Angle Class I (found in approximately 70% of subjects) and the least common malocclusion was Angle Class III (found in approximately 5% of subjects) [2,3,4]. In most of these investigations, researchers focused more on changes in the psychological well-being of the patients before and after orthodontic treatment. Refs. [5,6], Demirovic et al., in their study, concluded that dental malocclusion had a significant negative impact on oral health-related quality of life. Kara-Boulad et al. found that oral health-related quality of life improved in all groups after orthodontic treatment. Moreover, in a majority of the studies of patients wearing orthodontic appliances, the emphasis was on psychical pain or psychical discomfort. However, the psychological health of patients wearing orthodontic appliances during orthodontic treatment was not investigated. Thus, the null hypothesis of this study was that orthodontic appliances did not affect the well-being of the patients during orthodontic treatment.

Therefore, the purpose of this investigation was to analyze and evaluate the psychological health of patients wearing various orthodontic appliances during orthodontic treatment, emphasizing mental issues rather than psychical ones.

## 2. Materials and Methods

Investigation was performed using anonymous questionnaires made up of 31 questions that were composed to evaluate the psychological well-being of patients wearing orthodontic appliances. The authors confirm that all methods were carried out in accordance with relevant guidelines and regulations. All experimental protocols were approved by the Bioethical committee of the Lithuanian University of Health Sciences (No. BEC-OF-98, 21 February 2020). Informed consent was obtained from all subjects and/or their legal guardian(s).

Printed questionnaires in the Lithuanian language were given to the patients during their visit to their orthodontist. The questionnaire validation procedure was performed in two steps. First, the questions were evaluated by scientific experts in the current field for searching common errors, such as confusing or double questions. Later, the questionnaires were piloted (*n* = 25) for suitability for the main survey. The results of the pilot test met the expectations of the authors and were used for further investigation.

Questionnaire consisted of 7 questions about the demographical information of the patient; 13 questions about the psychological well-being of the patient; 3 questions about pain; 4 questions about eating difficulties; 3 questions about speech difficulties; 1 question about salivating. All of the questions were given with an option to choose from given answers or, if needed, to write their own. Investigation was performed in an Orthodontic Clinic. Selected subjects were patients wearing various orthodontic appliances and were randomly selected.

### Transparency and Openness

We report how we determined our sample size, all data exclusions (if any), all manipulations, and all measures in the study, and we follow JARS. Sample size was calculated using the Paniotto formula: *n* = 1/(∆^2 + 1/N). *n*—the sample size of subjects; ∆^2—the size of the sampling error; N—the general sample size. When the size of the sampling error was 0.05, the calculated sample size was 333. There were 339 patients in this investigation; therefore, the sample size of was sufficient.

The criteria for inclusion into investigation were as follows: patients that were being treated orthodontically in the Orthodontic Clinic without mental disorders and congenital syndromes.

All of the collected data were processed using the SPSS 22.0 (Statistical Package for Social Sciences) program which was used in collecting and analyzing data. Descriptive statistics were reported as the mean and standard deviation (SD). Hypotheses of interrelations between characteristics were verified using the Pearson chi-squared test (χ^2^). *p*-values less than 0.05 were considered significant. The power of the analysis was 0.8. This study’s design and its analysis were not pre-registered.

## 3. Results

This investigation included data of 339 patients. The average age of subjects was 14.08 years, the youngest patient was 7 years old and the oldest patient was 42 years old. As shown in the table below (Table 1), more than half of the investigated patients included were women (69%) and only 31% were men. Subjects were divided into four groups: (1) those younger than 14 years of age (14 year-olds included) wearing removable appliances; (2) those younger than 14 years of age wearing braces; (3) those older than 14 years of age wearing removable appliances; and (4) those older than 14 years of age wearing braces. This division was based on the average age of the subjects, which was around 14 years of age.

Between all age groups, reasons for seeking orthodontic treatment did not differ much. Both younger and older than 14 year-old patients indicated crooked teeth and bad bite as main reasons to seek orthodontic treatment (Table 2). However, women sought orthodontic treatment because of crooked teeth; meanwhile, men indicated bad bite as the main reason for treatment.

This investigation evaluated patients’ satisfaction with their smile before orthodontic treatment. It was discovered that 41.6% were satisfied with their smile while 48% were unsatisfied (Table 3). Interestingly, even though the number of participating women was much higher than the number of men, the satisfaction with their smile in women was lower.

Considering the psychological well-being of the patients, almost 50% of the subjects (48.7%) noted that they were feeling good and about 38% noted that they were feeling normal. Nevertheless, only 4% of all subjects included in this investigation noted that they were feeling bad while wearing orthodontic appliances. Also, a statistically significant difference was found between genders as women were more likely to feel very comfortable (great) while men usually felt good or normal while wearing orthodontic appliances. Results are shown in Table 4.

Only 9% of the subjects included in this investigation stated that they felt unhappy while wearing orthodontic appliances; however, 45% of subjects stated that they felt unhappy at least sometimes. It was found that patients younger than 14 years old wearing braces felt unhappy statistically more often (Table 5).

It was found that 43% of subjects felt stress while wearing orthodontic appliances. There was a statistically significant difference between patients younger than 14 years old wearing removable appliances and patients older than 14 years old wearing braces, as these two groups felt stress more often than other patient groups (Table 6).

Evaluation of eating difficulties with orthodontic appliances showed that 75% of patients had these difficulties while wearing braces. These patients had to stop eating certain kinds of foods such as nuts, chewing gum and popcorns to avoid any breakage of braces. When evaluating the effect of orthodontic appliances on structures of the oral cavity, it was found that 56% of the subjects had problems only in the beginning of the treatment and less than 20% of the subjects had these issues during the whole treatment. Also, statistically significantly more often, problems with mucosa were had by patients wearing braces and those who were men.

Speech difficulties were experienced by more than 30% of the subjects and about 26% indicated that they felt they experienced speech difficulties at least partially. Statistically significant differences in speech difficulties were experienced by patients wearing removable appliances. The most common speech difficulties were slurred speech (84%) and difficulties pronouncing certain words (16%).

Increased salivation was experienced by more than half of the subjects (58.8%). Statistically significantly more often, increased salivation was experienced by patients wearing removable appliances and more often by men. Moreover, subjects were asked whether they had to give up any leisure activities due to orthodontic treatment and only three respondents said they gave up playing wind instruments. Others did not indicate any restrictions of leisure activities due to orthodontic treatment.

## 4. Discussion

Usually, orthodontic treatment is performed for young patients because for the achievement of good results, it is necessary to start treatment in the still-growing patient (at least until the age of 14 years of age). However, orthodontic treatment can be proposed for older patients, but they require different methods of orthodontic treatment.

The main goal of orthodontic treatment is to balance esthetic and functional needs and patients’ ambitions, which contributes to their quality of life [5,6].

Several factors affect the quality of life during orthodontic treatment, such as pain, difficulties while eating, and changes in speech and diet [6,7]. Previous research had identified that fixed appliances affect everyday life, in terms of esthetics, functional limitations, diet, oral hygiene and socially [8,9]. Oral health status and quality of life were negatively affected during treatment, but improved afterwards [8,10,11]. Pain from fixed appliances reduced after a few days [8,12,13]. However, there were no studies on removable orthodontic appliances and retainers, although the ability to remove one’s appliance during eating, cleaning and talking could result in a different effect [8]. Thus, this investigation focused on the well-being of patients wearing removable and non-removable orthodontic appliances.

Of the 339 patients included in this study, more than two thirds were woman and only one third were men. Furthermore, the main reasons for women to seek orthodontic treatment were crooked teeth and esthetic reasons, while, for the majority of men, the main reason was a bad, traumatic bite. A study by Jung on Korean middle school-attending adolescents unveiled that after fixed orthodontic treatment, girls had higher self-esteem than the untreated malocclusion group compared to boys [14]. This showed that women tended to pay more attention to esthetics, thus, their self-esteem depended on their smile esthetics.

More than 50% of the subjects wearing orthodontic appliances evaluated their well-being as average and only 4% said they felt very bad. Moreover, patients wearing braces evaluated their well-being as normal or bad when compared to patients wearing removable appliances. This was because braces, which were fixed to a person’s teeth, could cause pain or discomfort, and it might make everyday life more difficult and result in feeling that they are ‘embarrassing’ or ‘upsetting’ [8]. Furthermore, younger than 14 year-old patients wearing removable appliances were less likely to feel great or good, compared to older than 14 year-old patients wearing removable appliances. This can be explained by the disturbances of speech, tongue irritation and difficulty in chewing with more social embarrassment for younger patients [15]. Whereas, older patients usually use removable appliances only at night such as retainers after orthodontic treatment.

It was discovered that only 9% of the subjects marked that they felt unhappy wearing orthodontic appliances; however, 45% of the subjects stated that they felt unhappy at least sometimes. Moreover, women were statistically significantly more prone to feel unhappy compared to men. Similar results were found in a study performed in 2020 in India. Authors discovered that there was a statistically significant difference between the overall psychological impact of dental esthetics and specific psychological impacts on male and female subjects, with female subjects having higher scores [15]. This may be explained by the fact that females were more concerned and dissatisfied with their dental appearance as compared to males, therefore, orthodontic treatment worsened women’s appearance for the time of the treatment and they felt more unhappy.

The majority of patients feeling stress during orthodontic treatment were patients wearing braces in both age groups. However, the majority of patients who did not feel stress were patients older than 14 years old wearing a removable appliance. This seems like a paradox as removable appliances usually affect speech and older patients wearing removable appliances should feel more stress, yet most of these patients wore removable appliances such as retainers after orthodontic treatment only at night; therefore, they did not feel stressed about it.

Furthermore, patients’ well-being greatly depended on discomfort or difficulties that occurred because of orthodontic treatment. Discomfort included speech difficulties, changes in eating habits, effects on oral structures, increased salivation and more attention paid to one’s teeth by others. The greater the discomfort, the lesser the well-being of the patient. Similar results were found in various studies around the world. An investigation carried out in the United Kingdom showed that 47% of students of 11–12 years of age experienced bullying due to the appearance of their teeth [16], thus, they evaluated their well-being poorly In our investigation, patients with braces had more difficulties compared to patients with removable appliances and this can be explained as patients with removable appliances removed them before eating while patients wearing braces could not do that. Moreover, in our investigation, 26% of the patients experienced at least partial difficulties with speech. Similar results were found in the study conducted by A. Kavaliauskiene and colleagues, where 26.8% of the subjects experienced speech difficulties [17]. Usually, removable appliances cover the palate and are relatively huge appliances, thus, the tongue has less space for articulation and difficulties in speech may appear. Furthermore, patients wearing removable appliances experienced increased salivation statistically significantly more often [18].

## 5. Conclusions

In conclusion, even though patients indicated various negative aspects of orthodontic appliances on their well-being during orthodontic treatment, for the majority of the subjects, the end results of the treatment significantly increased their psychological well-being, self esteem and social life].

### Limitations of the Study

The main limitation of this study was that the number of participants younger than 14 years old wearing braces and the number of patients older than 14 years old wearing removable appliances was low. Thus, the results of these groups should be supported by a higher number of participants in future investigations, paying attention to the homogeneity of the groups according to age, gender and the type of appliance.

## Figures and Tables

**Table 1 medicina-60-01287-t001:** Social and demographical characteristics of the patients (W—women, M—men).

Characteristics		*n*	Age	Percentage
Patients’ group	Younger than 14 years old wearing removable orthodontic plate	167 (116 W, 51 M)	9.3 ±1.23	49.3%
	Younger than 14 years old wearing braces	57 (39 W, 18 M)	10.3 ± 1.56	16.8%
	Older than 14 years old wearing removable orthodontic plate	8 (5 W, 3 M)	14.4 ± 0.95	2.4%
	Older than 14 years old wearing braces	107 (74 W, 31 M)	15.8 ± 1.12	31.6%
Gender	Men	105	12.8 ± 1.76	31.0%
	Women	234	11.7 ± 1.59	69.0%

**Table 2 medicina-60-01287-t002:** Reasons to seek orthodontic treatment in different patient groups.

		Crooked Teeth	Bad, Traumatic Bite	Esthetics	Other
Patient group	Younger than 14 years old wearing removable appliances	64.1%	48.5%	18.0%	0.0%
	Younger than 14 years old wearing braces	84.2%	52.6%	22.8%	0.0%
	Older than 14 years old wearing removable appliances	50.0%	0.0%	0.0%	50.0%
	Older than 14 years old wearing braces	55.1%	52.3%	37.4%	0.0%
	χ^2^	14.472	8.469	16.149	167.476
	df	3	3	3	3
	*p*	0.002 *	0.037 *	0.001 *	0.000 *
Gender	Men	55.2%	69.5%	11.4%	0.0%
	Women	68.4%	40.2%	30.3%	1.7%
	χ^2^	5.450	24.984	14.022	1.816
	df	1	1	1	1
	*p*	0.020 *	0.000 *	0.000 *	0.178

* Statistical significance when *p* < 0.05.

**Table 3 medicina-60-01287-t003:** Prevalence of satisfaction before orthodontic treatment in different patient groups.

		Satisfaction with Their Smile				χ^2^	df	*p*
		Very Satisfied	Satisfied	Unsatisfied	Very Unsatisfied			
Patient group	Younger than 14 years old wearing removable appliances	2.4%	44.3%	50.9%	2.4%	39.497	9	0.000 *
	Younger than 14 years old wearing braces	26.3%	24.6%	43.9%	5.3%			
	Older than 14 years old wearing removable appliances	0.0%	50.0%	50.0%	0.0%			
	Older than 14 years old wearing braces	15.0%	45.8%	39.3%	0.0%			
Gender	Men	3.8%	46.7%	49.5%	0.0%	10.742	3	0.013 *
	Women	13.2%	39.3%	44.4%	3.0%			

* Statistical significance when *p* < 0.05.

**Table 4 medicina-60-01287-t004:** Psychological well-being of patients wearing orthodontic appliances.

		Psychological Well-Being of Patients Wearing Orthodontic Appliances					χ^2^	df	*p*
		Great	Good	Normal	Bad	Very Bad			
Patient group	Younger than 14 years old wearing removable appliances	2.4%	56.3%	39.5%	1.8%	0.0%	73.398	9	0.000 *
	Younger than 14 years old wearing braces	1.8%	63.2%	24.6%	10.5%	0.0%			
	Older than 14 years old wearing removable appliances	50.0%	50.0%	0.0%	0.0%	0.0%			
	Older than 14 years old wearing braces	21.5%	29.0%	46.7%	2.8%	0.0%			
Gender	Men	0.0%	55.2%	41.9%	2.9%	0.0%	16.408	3	0.001 *
	Women	13.7%	45.7%	36.8%	3.8%	0.0%			

* Statistical significance when *p* < 0.05.

**Table 5 medicina-60-01287-t005:** Prevalence of feeling unhappy while wearing orthodontic appliances in different patient groups.

		Feeling Unhappy Wearing Orthodontic Appliances			χ^2^	df	*p*
		Yes	No	Sometimes			
Patient group	Younger than 14 years old wearing removable appliances	10.5%	56.1%	33.3%	20.804	6	0.002 *
	Younger than 14 years old wearing braces	7.8%	37.1%	55.1%			
	Older than 14 years old wearing removable appliances	0.0%	100.0%	0.0%			
	Older than 14 years old wearing braces	9.3%	51.4%	39.3%			

* Statistical significance when *p* < 0.05.

**Table 6 medicina-60-01287-t006:** Stress felt while wearing orthodontic appliances.

		Stress Felt			χ^2^	df	*p*
		Yes	No	Sometimes			
Patient group	Younger than 14 years old wearing removable appliances	9.0%	55.7%	35.3%	14.432	6	0.025 *
	Younger than 14 years old wearing braces	10.5%	64.9%	24.6%			
	Older than 14 years old wearing removable appliances	0.0%	100.0%	0.0%			
	Older than 14 years old wearing braces	18.7%	51.4%	29.9%			
Gender	Men	9.5%	50.5%	40.0%	5.946	2	0.051
	Women	13.2%	59.8%	26.9%			

* Statistical significance when *p* < 0.05.

## Data Availability

The datasets used and analyzed during the current study are available from the corresponding author on reasonable request.

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
