# Peer review of "Well-Being of Orthodontic Patients Wearing Orthodontic Appliances"

_medicina, 2024, doi:10.3390/medicina60081287_

Round 1

Reviewer 1 Report

Comments and Suggestions for Authors

Dear Authors,

The topic you have chosen to investigate is interesting and potentially beneficial. However, there are several major issues that need to be addressed:

It is unclear how you selected the patients (randomly, by time frame, etc.).

The number of participants in the groups varies widely, which could lead to questionable results.

Groups should be well-balanced by age, gender, and type of appliance, as some of the questions are directly related to the type of appliance.

The results appear to overemphasize women's perceptions (e.g., more than half of the investigated patients were women (69%) and only 31% were men). This should be presented as factual information, similar to what you did in line 94: "...69% women and 31% men."

Table 1 should include information about the groups, including type of appliance, gender, and age.

Lines 161-163 are not supported by the article you referred to, nor the original article, as the research was exclusively conducted on female patients in the original study.

No limitations of the study are mentioned.

Author Response

Dear reviewer,

Thank you for your valuable comments. Please find below the answers to your remarks:

1. The topic you have chosen to investigate is interesting and potentially beneficial. However, there are several major issues that need to be addressed:

It is unclear how you selected the patients (randomly, by time frame, etc.).

The  response. Thank you for the remark. We made changes:

Selected subjects were patients wearing various orthodontic appliances and were randomly selected.

2.The number of participants in the groups varies widely, which could lead to questionable results.

The  response. Agree. We mentioned this problem in the limitation section:

The main limitation of this study was that number of participants younger than 14 years old wearing braces and patients older than 14 years wearing removable appliances was low. Thus, the results of these groups should be supported by a higher number of participants in the future investigations, paying attention to homogeneity of the groups according to age, gender and type of appliance.

3. Groups should be well-balanced by age, gender, and type of appliance, as some of the questions are directly related to the type of appliance.

The  response. Thank you, fully agree. We mentioned this problem in the limitation section:

The main limitation of this study was that number of participants younger than 14 years old wearing braces and patients older than 14 years wearing removable appliances was low. Thus, the results of these groups should be supported by a higher number of participants in the future investigations, paying attention to homogeneity of the groups according to age, gender and type of appliance.

4.The results appear to overemphasize women's perceptions (e.g., more than half of the investigated patients were women (69%) and only 31% were men). This should be presented as factual information, similar to what you did in line 94: "...69% women and 31% men."

The  response. Thank you for the remark. We added additional information on this topic.

.....Interestingly, even though the number of participating women was much higher than men, the satisfaction with their smile in women was lower.

5.Table 1 should include information about the groups, including type of appliance, gender, and age.

The  response. Thank you for the point. We made changes in the Table 1 according to your remarks.

Characteristics

n

age

Percentage

Patients’ group

Younger than 14 years old wearing removable orthodontic plate

167 (116W,51M)

9.3 ±1.23

49.3%

Younger than 14 years old wearing braces

57( 39W,18M)

10.3±1.56

16.8%

Older than 14 years old wearing removable orthodontic plate

8( 5W, 3M)

14.4±0.95

2.4%

Older than 14 years old wearing braces

107( 74 W, 31M)

15.8±1.12

31.6%

Gender

Men

105

12.8±1.76

31.0%

Women

234

11.7±1.59

69.0%

  1. Lines 161-163 are not supported by the article you referred to, nor the original article, as the research was exclusively conducted on female patients in the original study.

The  response. Agree, our mistake. Thank you. We made changes in the reference list:

Jung MH (2010). Evaluation of the effects of malocclusion and orthodontic treatment on self-esteem in an adolescent population. American Journal of Orthodontics and Dentofacial Orthopedics; 138:160– 166.

7. No limitations of the study are mentioned.

The  response. Thank you for your valuable comments. We made additional section.

Limitations of the study

The main limitation of this study was that number of participants younger than 14 years old wearing braces and patients older than 14 years wearing removable appliances was low. Thus, the results of these groups should be supported by a higher number of participants in the future investigations, paying attention to homogeneity of the groups according to age, gender and type of appliance.

Reviewer 2 Report

Comments and Suggestions for Authors

Introduction

The purpose of the study is clearly stated, focusing on assessing the psychological health of patients wearing orthodontic appliances during treatment. However, the introduction contains some redundancies and could benefit from more transparent, more concise language. The lack of specificity in previous research, with no specific studies or examples mentioned, weakens the background context. The lack of hypotheses or research questions makes the direction of the study unclear. A lack of contextual details about the importance of mental health issues in orthodontic treatment is also noted. More detailed citations would direct readers to specific supporting studies.

Materials and methods

The section clearly describes the use of a detailed questionnaire by reporting a reasonable justification of the sample size, clear inclusion criteria, and statistical analysis. However, there needs to be a mention of whether the questionnaire was validated, which is crucial to ensure the reliability and validity of the data. Details on how the data were collected (e.g., online) must be included, affecting reproducibility and methodological understanding. The study design and analysis have yet to be pre-registered, which may raise concerns about transparency and potential bias. Potential confounding variables, such as previous orthodontic experience, are not addressed, which is essential for the robustness of the conclusions.

Results and Discussion 

The results describe the comprehensive treatment of various aspects of orthodontic treatment, such as age considerations, gender differences, psychological well-being, and functional difficulties, providing a holistic view of the topic. The discussion effectively compares the findings with those of previous research. In addition, statistically significant results are presented, adding credibility to the findings. Furthermore, there needs to be a discussion of the study's limitations, which is crucial to understanding the scope and applicability of the findings. There are no suggestions for future research or improvement of the patient experience.

Author Response

Dear reviewer,

Thank you  for your valuable comments. Please find  below the answers to your remarks:

  1. Introduction

The purpose of the study is clearly stated, focusing on assessing the psychological health of patients wearing orthodontic appliances during treatment. However, the introduction contains some redundancies and could benefit from more transparent, more concise language. The lack of specificity in previous research, with no specific studies or examples mentioned, weakens the background context. The lack of hypotheses or research questions makes the direction of the study unclear. A lack of contextual details about the importance of mental health issues in orthodontic treatment is also noted. More detailed citations would direct readers to specific supporting studies.

The response. Thank you for your valuable  comments. We  added additional information according to your recommendations.

In most of these investigations, researchers focused more on changes in psychological well-being of the patients before and after orthodontic treatment. [5-6]. Demirovic et all. in their study concluded that dental malocclusion had significant negative impact on oral health-related quality of life .Kara-Boulad et all. found that oral health-related quality of life improved in all groups after orthodontic treatment. Moreover, in a majority of the researches of patients wearing orthodontic appliances emphasis was on psychical pain or psychical discomfort. However, psychological health of patients wearing orthodontic appliances during orthodontic treatment was not investigated. Thus, the null hypothesis of this study was that orthodontic appliances did not affect the well-being of the patients during orthodontic treatment.

  1. Materials and methods

The section clearly describes the use of a detailed questionnaire by reporting a reasonable justification of the sample size, clear inclusion criteria, and statistical analysis. However, there needs to be a mention of whether the questionnaire was validated, which is crucial to ensure the reliability and validity of the data. Details on how the data were collected (e.g., online) must be included, affecting reproducibility and methodological understanding. The study design and analysis have yet to be pre-registered, which may raise concerns about transparency and potential bias. Potential confounding variables, such as previous orthodontic experience, are not addressed, which is essential for the robustness of the conclusions.

The response. Thank you for your valuable  comments. We  added additional information according to your recommendations.

Printed questionnaires in Lithuanian language were given to the patients during their visit to orthodontist. The questionnaire validation procedure was performed in two steps. First of all, the questions were evaluated by the scientific experts on the current field for searching common errors, such as confusing or double questions. Later, the questionnaires were piloted (n=25) for suitability for the main survey. The results of pilot test  met the expectations of the authors and were used for further investigation.

3.Results and Discussion 

The results describe the comprehensive treatment of various aspects of orthodontic treatment, such as age considerations, gender differences, psychological well-being, and functional difficulties, providing a holistic view of the topic. The discussion effectively compares the findings with those of previous research. In addition, statistically significant results are presented, adding credibility to the findings. Furthermore, there needs to be a discussion of the study's limitations, which is crucial to understanding the scope and applicability of the findings. There are no suggestions for future research or improvement of the patient experience.

The response. Thank you for your valuable  comments. We  added additional information according to your recommendations.

Limitations of the study

The main limitation of this study was that number of participants younger than 14 years old wearing braces and patients older than 14 years wearing removable appliances was low. Thus, the results of these groups should be supported by a higher number of participants in the future investigations, paying attention to homogeneity of the groups according to age, gender and type of appliance.

Round 2

Reviewer 1 Report

Comments and Suggestions for Authors

Thank you for accepting suggestions. Please continue with this research, as it is very beneficial to both orthodontists and patients. And always be aware of bias :)

Reviewer 2 Report

Comments and Suggestions for Authors

The Authors have correctly included all the critical points that I had highlighted in the previous report. Therefore, the manuscript can be positively considered for publication in this journal.